# An Experimental Apparatus for E-Nose Breath Analysis in Respiratory Failure Patients

**DOI:** 10.3390/diagnostics12040776

**Published:** 2022-03-22

**Authors:** Carmen Bax, Stefano Robbiani, Emanuela Zannin, Laura Capelli, Christian Ratti, Simone Bonetti, Luca Novelli, Federico Raimondi, Fabiano Di Marco, Raffaele L. Dellacà

**Affiliations:** 1Department of Chemistry, Materials and Chemical Engineering “Giulio Natta” (DCMC), Politecnico di Milano, 20133 Milano, Italy; carmen.bax@polimi.it (C.B.); christian.ratti@mail.polimi.it (C.R.); 2TechRes Lab, Department of Electronics Information and Bioengineering (DEIB), Politecnico di Milano, 20133 Milano, Italy; stefano.robbiani@polimi.it (S.R.); emanuela.zannin@polimi.it (E.Z.); raffaele.dellaca@polimi.it (R.L.D.); 3Unit of Pneumology, Azienda Ospedaliera Socio Sanitaria Territoriale Papa Giovanni XXIII, 24127 Bergamo, Italy; simone.bonetti@unimi.it (S.B.); lnovelli@asst-pg23.it (L.N.); fraimondi@asst-pg23.it (F.R.); fdimarco@asst-pg23.it (F.D.M.); 4Department of Health Sciences, Università degli Studi di Milano, 20142 Milano, Italy

**Keywords:** COVID-19, electronic nose, breath analysis, odour analysis, diagnosis

## Abstract

Background: Non-invasive, bedside diagnostic tools are extremely important for tailo ring the management of respiratory failure patients. The use of electronic noses (ENs) for exhaled breath analysis has the potential to provide useful information for phenotyping different respiratory disorders and improving diagnosis, but their application in respiratory failure patients remains a challenge. We developed a novel measurement apparatus for analysing exhaled breath in such patients. Methods: The breath sampling apparatus uses hospital medical air and oxygen pipeline systems to control the fraction of inspired oxygen and prevent contamination of exhaled gas from ambient Volatile Organic Compounds (VOCs) It is designed to minimise the dead space and respiratory load imposed on patients. Breath odour fingerprints were assessed using a commercial EN with custom MOX sensors. We carried out a feasibility study on 33 SARS-CoV-2 patients (25 with respiratory failure and 8 asymptomatic) and 22 controls to gather data on tolerability and for a preliminary assessment of sensitivity and specificity. The most significant features for the discrimination between breath-odour fingerprints from respiratory failure patients and controls were identified using the Boruta algorithm and then implemented in the development of a support vector machine (SVM) classification model. Results: The novel sampling system was well-tolerated by all patients. The SVM differentiated between respiratory failure patients and controls with an accuracy of 0.81 (area under the ROC curve) and a sensitivity and specificity of 0.920 and 0.682, respectively. The selected features were significantly different in SARS-CoV-2 patients with respiratory failure versus controls and asymptomatic SARS-CoV-2 patients (*p* < 0.001 and 0.046, respectively). Conclusions: the developed system is suitable for the collection of exhaled breath samples from respiratory failure patients. Our preliminary results suggest that breath-odour fingerprints may be sensitive markers of lung disease severity and aetiology.

## 1. Introduction

Respiratory failure is a severe condition requiring prompt medical intervention; it may have different origins, including airway obstruction, infections, interstitial lung disease, pulmonary oedema, etc. Once a patient is diagnosed with respiratory failure, the underlying cause must be effectively treated as soon as possible. Different clinical tests are necessary to identify the cause of respiratory failure and guide treatments [1], including chest X-ray, chest CT, bronchoscopy, bacterial cultures, and lung biopsy. Some of these tests are invasive, expose the patient to radiation risks, require a long time before the results are available, and use sophisticated and cumbersome equipment. Novel, simple, bedside, non-invasive, and fast diagnostic methods capable of identifying the role of infection and/or inflammation in respiratory failure are needed to improve monitoring its progression, guiding treatments and contributing to the appropriate use of antibiotics and steroids [2].

Exhaled breath contains >3000 Volatile Organic Compounds (VOCs) resulting from physiological and pathological metabolic processes [3,4,5] that can be measured non-invasively. Indeed, there are several scientific studies discussing the possibility of diagnosing disparate diseases from the analysis and identification of volatile biomarkers in exhaled breath by means of different analytical techniques [6,7]. Most studies focus on the non-invasive detection of cancer [8,9,10], but there are also other examples regarding tuberculosis [11] and liver cirrhosis [12,13,14]. Despite the relatively high number of VOCs claimed as potential biomarkers, only a few compounds have been effectively demonstrated and approved for clinical applications. This may be related to different reasons, including large individual variations in the biomarker concentrations in diseased and/or healthy subjects [15]. In this context, the electronic nose (EN) represents an emerging non-invasive technique employing a combination of low-cost gas sensor arrays and pattern recognition algorithms to detect and differentiate VOC patterns (known also as “odour fingerprints”) [16,17] without requiring the identification of single chemical compounds. Compared with other analytical techniques, ENs have the advantages of being generally more flexible, smaller, less expensive, and easier to use and having rapid response and quick sensor recovery times [18], making them more appealing for point-of-care applications. Some disadvantages of ENs as diagnostic devices are the poor reproducibility of gas sensors and the relatively short sensor life due to drift [19]. In more detail, because of the poor reproducibility of gas sensors, prediction models developed on an instrument cannot be transferred to other devices, but a recalibration phase is needed [20,21]. Sensor drift leads to a progressive worsening of classification performances over time, resulting in the need for periodical recalibrations of the system to update the classification models [18]. Consequently, the scaling of the EN to an industrial level results in an expensive and time-intensive process [20].

The analysis of exhaled breath by ENs has gained momentum in recent years thanks to the advances in sensor technologies and data analytics [22]. In particular, ENs were proven to be sensitive to various factors, such as airway inflammation [23] and viral or bacterial infections [24,25]. Moreover, ENs were proven to be capable of identifying and classifying various bacterial species cultured in-vitro [26,27] and bowel infections [28]. Recently, evidence supporting the applicability of EN technology for the detection of COVID-19 has been reported in the scientific literature [29,30].

In general, diagnostic systems for the analysis of VOCs in exhaled breath should be able to filter out or compensate for ambient VOCs; this is crucial in critical care units, where ambient air may contain VOCs exhaled by patients with different diseases or released by cleaning chemicals and drugs. As reported in the scientific literature, this aspect has been addressed in different ways. Some research groups equipped the EN with a charcoal filter to filter ambient air during lung washout to prevent breath sample contamination [31]. Others compensated for ambient VOCs by subtracting the responses to blank samples from the EN responses to breath samples, analysed in parallel using nominally identical sensors [32]. Nevertheless, this approach is based on the assumption that nominally identical sensors respond in the same way as the same stimulus, thus neglecting the well-known problem of poor reproducibility of gas sensors.

Moreover, breath sampling systems suitable for patients with respiratory failure should impose a low additional mechanical breathing load to the patient and be able to deliver breathing gas mixtures at different oxygen concentrations. To our knowledge, none of the currently available breath analysers or sampling systems described in the literature to collect exhaled breath for VOC analysis by ENs satisfies these requirements.

In the present study, we aimed to develop a novel experimental set-up suitable for sampling exhaled breath at the bedside in patients with acute respiratory failure. Furthermore, we ran a feasibility study using this novel system in (1) patients with respiratory failure due to SARS-CoV-2, (2) patients with SARS-CoV-2 infection without respiratory failure, and (3) controls [33,34,35,36]. In a preliminary analysis, we evaluated whether our system could differentiate respiratory failure patients from controls. In a further exploratory analysis, we investigated whether our system was sensitive to the severity of lung disease in SARS-CoV-2 patients with and without respiratory failure.

## 2. Materials and Methods

### 2.1. Setup for Exhaled Breath Collection

The main goal of the present study was the development of an experimental apparatus for exhaled breath sampling specifically designed for patients affected by respiratory failure. A critical aspect of breath analysis is that ambient air usually contains VOCs, which alter the composition of exhaled breath and thus may affect the diagnosis. Lung washout with clean air can reduce such contamination [37] and is commonly achieved by making the subject inhale through a charcoal filter [38]. However, such a procedure is not suitable for patients with respiratory failure because the high resistance of charcoal filters would increase the work of breathing beyond the patient’s capability. Moreover, such patients often require oxygen-enriched gasses. Therefore, we developed an apparatus that allows lung washout [37] using medical grade air and oxygen that does not increase the mechanical load perceived by the patient.

The proposed apparatus comprises a T-piece to allow a bias flow through the breathing circuit, a non-rebreathing valve (BB089YBPV, Burke & Burke SPA, Milan, Italy), and an antiviral filter to avoid cross-contamination (Figure 1). The system is connected to the hospital medical gas pipeline system to provide clean gas to prevent the possible influence of ambient VOCs. The connection to the medical gas pipeline system also allows the regulation of the fraction of inspired oxygen (FiO_2_) according to the patient’s needs. The amount of bias flow of medical gasses passing through the breathing circuit, as well as the FiO_2_, can be adjusted using two rotameters on the air and oxygen lines. To determine the value of the settings for medical air and oxygen, we used the following criteria: (1) the sum of medical air and oxygen flow must be greater than the peak inspiratory flow of the patient and (2) the flow of oxygen is determined for providing the desired FiO_2_. In the present study, we used a bias flow of 45 L/min and an FiO_2_ of 0.21. A 10 cm-long tube at the outlet of the T-piece guarantees that the subject inhales only the gas from the pipeline system and not the environment air. Exhaled breath is collected into Nalophan^TM^ bags specific for olfactometric analyses (European Standard EN13725:2003) equipped with two Teflon^TM^ tubes, one on each side. One tube connects the bag to the breath sampling system during the test, and the other one is used to connect the bag to the EN for the analysis. The sampling procedure comprises two phases: (1) lung washout from ambient VOCs and (2) exhaled breath sampling. During the washout phase, the patient inhales the mixture of fresh medical gasses through the check (non-return) valve in the inspiratory limb of the non-rebreathing valve and exhales into the ambient by the expiratory limb check valve. During the sampling phase, a Nalophan^TM^ bag with a capacity of 5 L is connected to the expiratory outlet of the non-rebreathing valve to collect exhaled gas.

### 2.2. Feasibility Study

A feasibility study was conducted at the ASST Papa Giovanni XXIII Hospital, Bergamo, Italy. The local ethical committee approved the study on 15/09/2020 (approval number 223/20). Written informed consent was obtained from all subjects prior to recruitment. 

Clinicians asked the subjects to interrupt the measurement if they felt discomfort or dyspnoea, and arterial oxygen saturation (SpO_2_) was measured by pulse oximetry before, during, and after the measurement. A measurement was considered successful if the procedure (i.e., washout of both lungs and exhaled breath sampling) was completed without discomfort, dyspnoea, or desaturations (defined as SpO_2_ below 88%). The primary endpoint was the percentage of successful measurements.

#### 2.2.1. Study Population 

Aiming for 90% power at a significance level of 0.05 and hypothesising an area under the ROC curve (AUC) of 0.80 for the EN and 0.50 for the null hypothesis [39], we calculated a minimum required sample size of 17 subjects per group (respiratory failure vs. controls). We enrolled 55 subjects between March and September 2021 on a convenience basis. Subjects belonged to the following groups: (1) 25 patients with respiratory failure positive for SARS-CoV-2; (2) 8 asymptomatic subjects positive for SARS-CoV-2; and (3) 22 healthy controls. Infection with SARS-CoV-2 was determined on the basis of the result of a PCR test using oropharyngeal swabs. Controls were age-matched subjects with a negative molecular swab for SARS-CoV-2 without chronic respiratory disorders who had not presented with any respiratory tract infection in the last 90 days. Patients with respiratory failure were studied when they were deemed stable by the attending physician breathing room air for a few minutes. Chi-squared tests and ANOVA were used to investigate possible differences between groups. 

#### 2.2.2. Exhaled Breath Sampling

Subjects were asked not to eat, drink, or smoke in the two hours before the study. The exhaled breath sampling protocol consisted of two phases: (1) lung washout from ambient VOCs and (2) exhaled breath sampling.
Washout phase: The patient breathed for 3 min through the setup with the exhaled line of the non-rebreathing valve open, which allowed the inhalation of a mixture of medical gasses from the hospital pipeline system and exhalation into the ambient (Figure 1c). The subject wore a nose clip to avoid nasal respiration and breathed through a mouthpiece.Exhaled breath sampling: At the end of the washout phase, patients held their breath while the operator connected a Nalophan^TM^ bag to the expiratory line of the non-rebreathing valve to collect the exhaled breath (Figure 1d).

Each patient performed a single measurement. The sampling, requiring about 5 breaths, lasted about 5 min. The exhaled breath samples were stored for 2 to 24 h after collection in the same room where the EN was installed. This conditioning time allowed the reduction of the moisture content of the exhaled breath samples thanks to the Nalophan^TM^ permeability to humidity. Modulating sample humidity ensures good stability and reproducibility of EN responses [40]. Indeed, the high humidity of exhaled breath interferes with the adsorption of VOCs on the MOX sensors.

#### 2.2.3. Exhaled Breath Analysis 

The EN used to analyse the exhaled breath samples was a commercial instrument (EOS-AROMA, SACMI S.C.) equipped with 4 custom-made Metal Oxide (MOX) sensors based on different metal oxides (Figure 2) [40,41]. In more detail, we used one ZnO-based sensor, one TiO2-based sensor, and two SnO-based sensors, all operated at about 400 °C by the 5 V-powered Pt heater.

The EN analysis consisted of two phases:Adsorption: the exhaled breath samples were drawn at a constant flow rate of 50 mL/min for 20 min into the sensor chamber using a vacuum pump. In this phase, a decrease in the MOX sensor resistance was recorded.Desorption: room reference air was drawn into the sensor chamber at a constant flow rate of 50 mL/min for 20 min to restore the sensor’s baseline. Each sample was analysed once with this instrumentation.

#### 2.2.4. Data Processing

The data processing procedure developed for this study comprised three steps: feature extraction, feature selection, and pattern recognition (Figure 3). Ten-fold cross-validation was carried out to optimise and validate the feature selection and pattern recognition models.

### 2.3. Feature Extraction

Various features representative of both steady-state and transient conditions were computed from EN sensor responses. A total of 112 features were considered. Table 1 reports the features considered for each sensor and the relevant evaluation time points. The resulting multi-dimensional feature set was auto-scaled.

### 2.4. Feature Selection

We applied the Boruta algorithm to identify the features that better discriminated between respiratory failure patients with SARS-CoV-2 and controls. Boruta is based on a wrapper approach built around a Random Forest classifier. Briefly, Boruta ranks features according to their information gain by evaluating the loss of classification accuracy caused by a random permutation of feature values between objects [41,42].

### 2.5. Pattern Recognition

The selected features were processed using Principal Component Analysis (PCA) [43], and PCA scores were used as the inputs of a Support Vector Machine (SVM) classifier [44]. The SVM searches for the hyperplane/function that enables the correct classification of most training data to define support vectors for future predictions. Support vectors define the margins of the hyperplanes, and the complexity of the classification task depends on their number. For non-linearly separable data in the input space, a Kernel trick should be used to map the data into a higher-dimensional space where they can be linearly separated. In the present study, we used the Gaussian radial basis kernel function (radial SVM): $$K\left(x,x^{\prime }\right)=\exp\left(-\|x-x^{\prime }{\|}^{2}/2\sigma^{2}\right)$$
where *σ* is a tuning parameter related to the model variance and $\|x-x^{\prime }{\|}^{2}$ is the squared Euclidean distance between two points [44].

### 2.6. Ten-Fold Cross-Validation

Ten-fold cross-validation was used to optimise feature selection and pattern recognition and estimate the performances of the proposed model. The original data set was partitioned into 10 subsets. Each subset was used once as a test set, while the others were used as training sets. The procedure was repeated 10 times, and the total performance was computed as the sum of the result on each fold [45].

### 2.7. Evaluation of Classification Capability

The results of PCA were reported in a two-dimensional score plot to allow for a visual assessment of the EN’s capability to discriminate the odour fingerprints of SARS-CoV-2 patients with respiratory failure versus controls. To investigate whether the classification model was sensitive to respiratory failure or SARS-CoV-2 infection, we superimposed the PC scores from asymptomatic SARS-CoV-2 patients on the score plot. We also computed the norm of the PC scores for each subject and compared the three groups using one-way ANOVA with the Holm–Sidak post hoc method. The diagnostic accuracy of the model represented was evaluated using the area under the Receiving Operating Characteristic (ROC) and expressed both as the area under the ROC curve (AUC), sensitivity, and specificity. 

## 3. Results

The measurement apparatus proved suitable for the critical care setting and well tolerated by the patients, and 100% of the measurements were completed successfully. 

### 3.1. Characteristics of Study Participants

A total of 55 subjects were recruited and tested in random order throughout the research: (1) 25 patients with positive SARS-CoV-2 swabs and respiratory failure, (2) 8 asymptomatic patients with positive SARS-CoV-2 swabs, and (3) 22 subjects with negative SARS-CoV-2 PCR tests on oropharyngeal swabs. The characteristics of study participants are summarised in Table 2. Patients with respiratory failure due to SARS-CoV-2 were studied after a median (IQR) of 7.5 (6.1, 10.3) days from the onset of symptoms. At the time of the study, 10 (40%) respiratory failure patients were receiving systemic steroids, 11 (44%) were receiving antibiotics, and 16 (64%) were receiving low-flow oxygen therapy with a median (IQR) fraction of inspired oxygen of 35% (30%, 36%). 

### 3.2. Features of Interest

Figure 4 illustrates typical EN responses resulting from the analysis of exhaled breath samples from a respiratory failure patient with SARS-CoV-2 and a control subject.

A preliminary visual inspection of EN sensor responses highlighted that samples from different groups, i.e., patients with respiratory failure due to SARS-CoV-2 and controls, interacted differently with the EN sensors, causing different resistance variation rates during the absorption and desorption phases. 

We extracted 112 features from the sensors’ responses, as described in Section 2.2.4. The 112-dimensional vectors were processed using the Boruta algorithm to reduce data dimensionality. Figure 5 illustrates the output of the Boruta features selection algorithm. The algorithm automatically selected 9 features out of 112, with an importance score lower than the shadow features’ maximum score. The selected features were related to the desorption phase of sensors 2 and 4. Such features were used for further processing.

### 3.3. Classification and Correlation of EN Responses with Respiratory Failure

In this feasibility study, we implemented a preliminary classification model to discriminate between respiratory failure patients with SARS-CoV-2 and controls. 

The first two principal components explained 92% of the variance. From the score plot relevant to these first two principal components (Figure 6a), it is possible to observe that respiratory failure patients with SARS-CoV-2 and controls were well separated. Specifically, samples from respiratory failure patients with SARS-CoV-2 clustered in the lower left part of the plot, while samples from controls clustered in the upper right part. This evidence suggests that the EN has the potential to identify the specific odour fingerprint associated with SARS-CoV-2 respiratory failure.

As an exploratory analysis, we superimposed the results from asymptomatic SARS-CoV-2 subjects on the score plot to investigate whether the classification model was sensitive simply to SARS-CoV-2 infection or its combination with respiratory failure. Interestingly, the asymptomatic SARS-CoV-2 subjects clustered between respiratory failure patients with SARS-CoV-2 and controls. 

Figure 6b shows the 2-Norm of the scores of all the principal components in the three groups. Asymptomatic SARS-CoV-2 patients fell between respiratory failure patients with SARS-CoV-2 and controls, suggesting that the EN may provide information about the severity of lung disease. The odour fingerprints of respiratory failure patients with SARS-CoV-2 were significantly different from those of controls (*p* < 0.001) and asymptomatic SARS-CoV-2 patients (*p* = 0.046). We found no significant differences between asymptomatic SARS-CoV-2 patients and controls (*p* = 0.028).

Figure 7 reports the Receiving Operating Characteristic (ROC) curve representing the EN accuracy in discriminating between respiratory failure patients with SARS-CoV-2 and controls. The area under the ROC curve was equal to 0.81 (CI_95%_ 0.74–0.88), with a sensitivity 0.920 (CI_95%_ 0.87–0.99) and a specificity of 0.682 (CI_95%_ 0.54–0.78).

## 4. Discussion and Conclusions

The assessment of VOCs in exhaled breath is a non-invasive bedside test that has the potential to provide clinically relevant information on patients with respiratory failure to improve the monitoring of the progression of disease severity and information on the effects of treatments. However, analysing the exhaled breath in patients with respiratory failure in the critical care setting is challenging, mainly because of the patients’ unstable conditions, but also because of the presence of VOCs in the environment. This study presents a new experimental setup to collect exhaled breath in patients with respiratory failure, which enables these challenges to be overcome. The setup makes use of medical gasses available from the hospital pipeline system to both wash out the lungs of ambient VOCs and collect uncontaminated exhaled breath without increasing the patient’s work of breathing. Such a solution also allows the investigator to provide to the subject an oxygen-enriched gas mixture with fine control of the FiO_2_, as is commonly required by respiratory failure patients. This system was evaluated in stable SARS-CoV-2 patients with and without respiratory failure and in healthy controls. The measurement apparatus and procedure proved suitable for the critical care environment. We did not record adverse events during the measurements, including desaturations, and the subjects did not report increased dyspnoea or discomfort. Since the oxygen concentration in the exhaled breath may interfere with the MOX sensor response, in the present study, we standardised the FiO_2_ level to 0.21. If patients required higher FiO_2_ values, we recommend using the same value for all subjects or applying a correction model that accounted for the different MOS sensor responses at different oxygen concentrations. 

We tested our system on a group of 55 subjects. The subjects included patients with respiratory failure due to SARS-CoV-2, asymptomatic patients with SARS-CoV-2, and healthy controls. We initially developed a classification model to investigate whether the device could detect differences in the odour fingerprints of patients with respiratory failure compared with controls. Since the sample size was not adequate to train and validate a robust and generalisable classification model, the performances of the classifier, obtained by cross-validation, should be considered a preliminary analysis to test the ability of the developed system to discriminate between markedly different classes. A sensitivity and specificity of 92% and 69%, respectively, were achieved in this feasibility study. 

Our results proved ENs’ potential to discriminate between SARS-CoV-2 patients with respiratory failure and a control group, thereby supporting further investigations in this innovative research field. 

We also performed an exploratory analysis that included asymptomatic patients who tested positive for SARS-CoV-2. This group was clearly distinguished from the control group, suggesting that the proposed method might be able to identify subjects positive for SARS-CoV-2 even if they do not present symptoms. Moreover, our preliminary data show that asymptomatic SARS-CoV-2 patients exhibit an intermediate response between that of SARS-CoV-2 symptomatic patients and controls. This finding is very interesting because it suggests that the proposed method is sensitive both to the presence of an infection and the severity of the patient’s condition. Further studies are needed to investigate whether the proposed method can be used to identify the aetiology of respiratory failure and monitor the progression of disease severity. 

The results achieved within this feasibility study also pointed out that the classification performance is comparable with those reported in the scientific literature. Indeed, other studies proposing the use of ENs as SARS-CoV-S screening tools achieved accuracies ranging from 0.86 to 0.95, sensitivities from 0.86 to 0.94, and specificities from 0.70 to 0.90 [29,46,47]. 

Compared with other devices based on the same measuring technology, e.g., Cyranose, Aeonose, GeNose, and SpiroNose, the novelty of the approach here presented concerns regarding the applicability of the developed breath sampling system to patients with respiratory failure. In more detail, the developed breath sampling system using medical grade air and oxygen, enables lung washout of environmental VOCs, ensuring the measurements’ reproducibility. Moreover, our system allows washout without increasing the dead space or the mechanical load perceived by the patient. In current commercial devices, washout is, in general, obtained by making the subject inhale through a charcoal filter, which adds respiratory load and is usually not certified to be used with oxygen-enriched mixtures. 

The SpiroNose uses a different approach: it compensates for ambient VOCs by using two sets of nominally identical sensors: the first exposed to exhaled breath and the second to environmental air. The results from the environmental sensors are then subtracted from those of the exhaled breath. Although this approach is potentially applicable in subjects with respiratory failure, actual systems cannot be used when patients are breathing oxygen-enriched gas mixtures.

In conclusion, we developed a novel system allowing the collection of exhaled breath in patients with respiratory failure for EN tests. The preliminary analysis performed on the collected breath samples shows that such a system could differentiate between patients with respiratory failure and controls and suggests that the response may be sensitive to the severity of the lung disease. The results of this feasibility study support further developments of this technology aimed at investigating relevant clinical applications of this approach in patients suffering from respiratory failure, such as the identification of disease aetiology and the monitoring of disease progression.

## Figures and Tables

**Figure 1 diagnostics-12-00776-f001:**
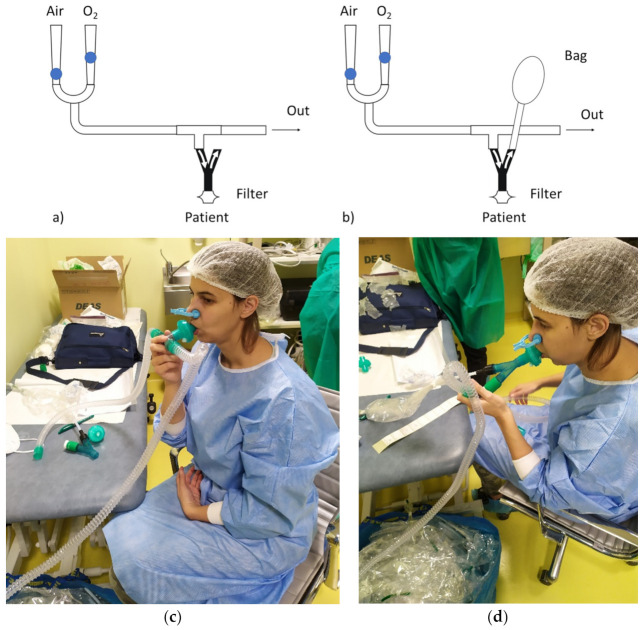
Proposed breath sampling apparatus. (**a**–**c**) Washout phase. The apparatus is connected to the medical gas pipeline system through two rotameters (one for medical air and the other one for oxygen), a tube, and a T connector. The patient is connected to the T connector through the inspiratory line of the non-rebreathing valve. (**b**–**d**) Exhaled breath sampling. The sampling bag is connected to the expiratory line of the non-rebreathing valve.

**Figure 2 diagnostics-12-00776-f002:**
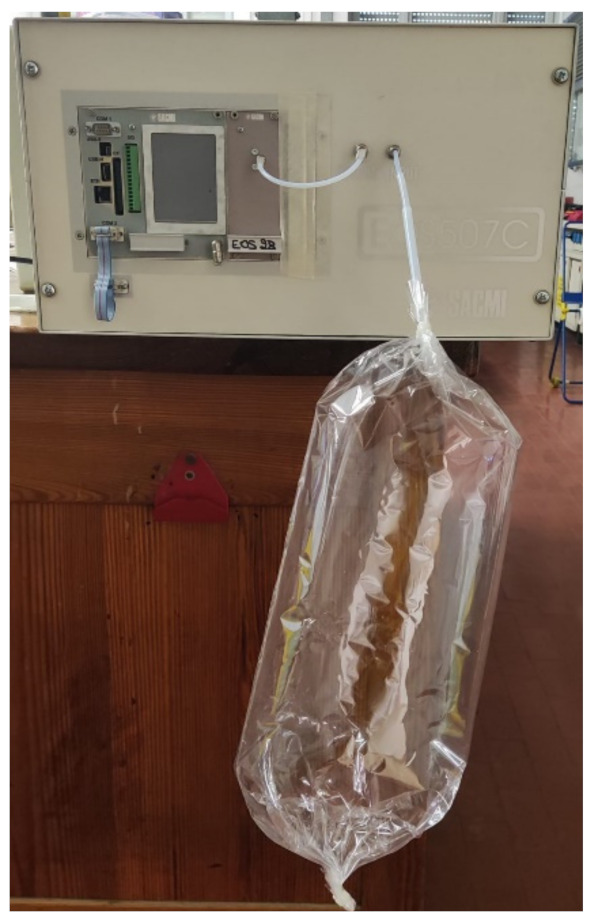
EN analysis of breath samples.

**Figure 3 diagnostics-12-00776-f003:**
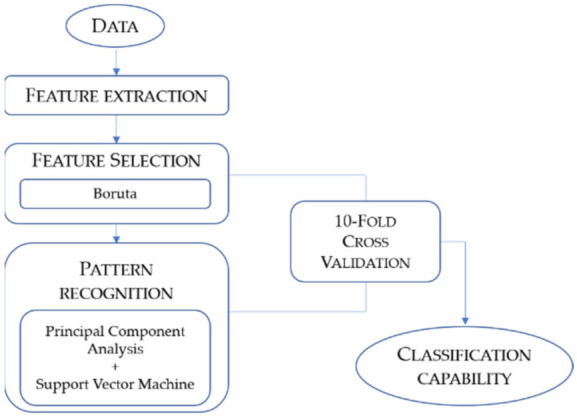
Schematic representation of the data processing procedure adopted for the study.

**Figure 4 diagnostics-12-00776-f004:**
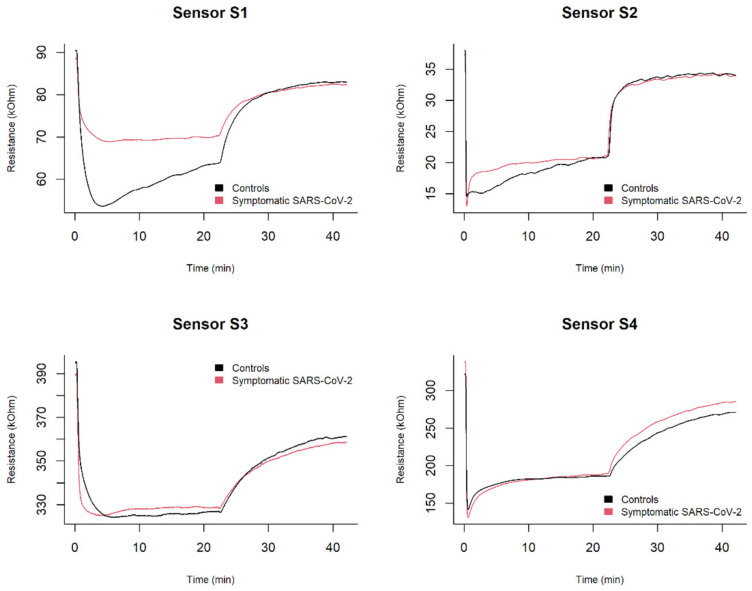
Representative EN responses to exhaled breath samples in a patient with respiratory failure due to SARS-CoV-2 (black) and in a control subject (red).

**Figure 5 diagnostics-12-00776-f005:**
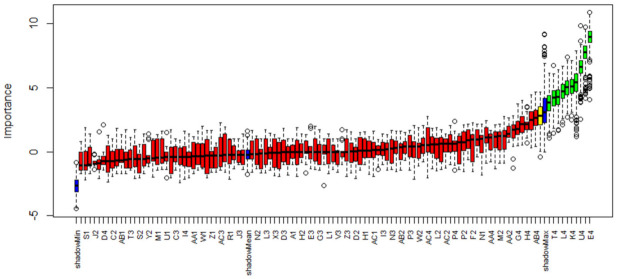
Output of the feature selection model based on the Boruta algorithm. In red: rejected features (i.e., variables useless for classification purposes); in yellow: tentative features (i.e., a variable whose importance was so close to their best shadow attributes that algorithm was not able to make a decision with the desired confidence in default number of random forest runs); in green: selected features (i.e., variables important for classification purposes); in blue: shadow features (i.e., duplicates of the original features, in which the values are randomly shuffled to eliminate the correlation between the variable values and the belonging class).

**Figure 6 diagnostics-12-00776-f006:**
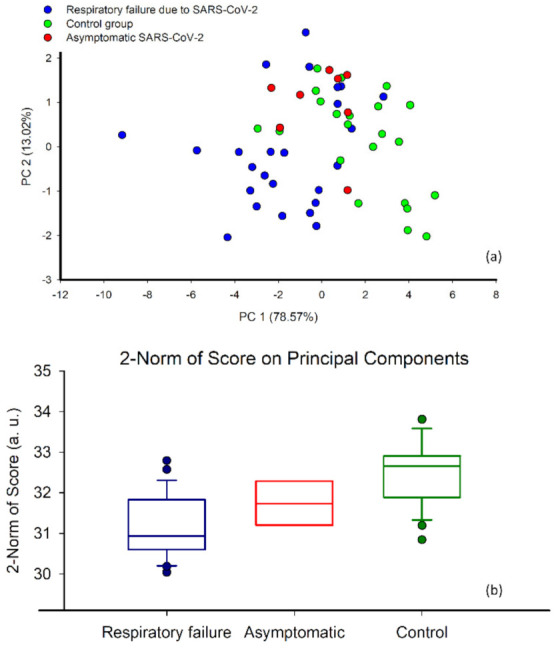
(**a**) Principal components of the selected features in asymptomatic SARS-CoV-2 patients (red circles), respiratory failure patients with SARS-CoV-2 (blue circles), and control subjects (green circles). (**b**) 2−Norm of scores of all the Principal Component of SARS-CoV-2 patients with respiratory failure, asymptomatic SARS-CoV-2 patients, and controls. The boundaries of the boxes indicate the 25th and 75th percentiles, and the lines within the boxes mark the median values. Whiskers define the 90th and 10th percentiles. Closed circles are the outliers.

**Figure 7 diagnostics-12-00776-f007:**
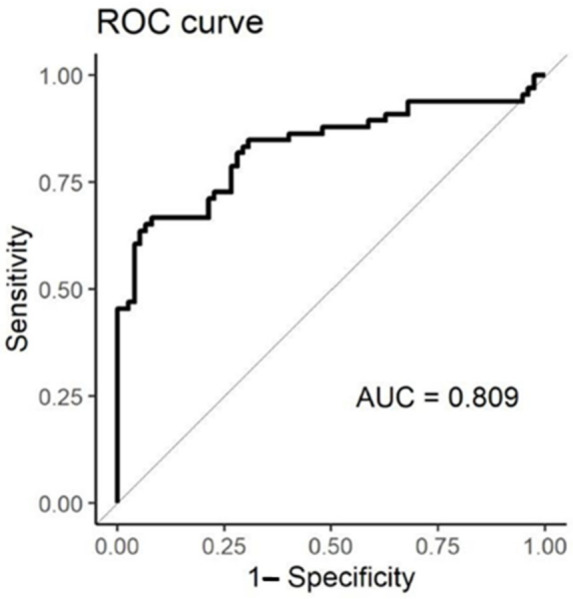
Receiving Operating Characteristic (ROC) curve representing the EN accuracy in discriminating between respiratory failure patients with SARS-CoV-2 and controls.

**Table 1 diagnostics-12-00776-t001:** Steady-state and transient features extracted from EN responses.

Feature	Description	Point of Evaluation
Resistance ratio	$\frac{R_{i}}{R_{0}}$	*i* = End of Adsorption and/or Desorption phase
Resistance difference	$R_{i}-R_{0}$	*i* = End of Adsorption and/or Desorption phase
Area under the curve	Area under the curve	Adsorption phase, Desorption phase, Overall analysis
Phase integral	Area under the plot x = R, y = dR/dt	Adsorption phase, Desorption phase, Overall analysis
Single point	$R_{i}$	Adsorption and Desorption phase *i* = Middle and End of Adsorption phase
Difference ratio	$\frac{R_{3}-R_{2}}{R_{2}-R_{1}}$	1 = Beginning of Adsorption phase 2 = Middle of Adsorption phase 3 = End of Adsorption phase
Last difference	*R*_3_ − *R*_2_	2 = Middle of Adsorption phase 3 = End of Adsorption phase
Exponential moving average	y_k_ = (1 − α)y_k−1_ + α(x_k_ − x_k−1_)	Max, Min, Area under peaks α = (0.01, 0.001)

**Table 2 diagnostics-12-00776-t002:** Characteristics of study participants.

	SARS-CoV-2 and Respiratory Failure	SARS-CoV-2 and Asymptomatic	Controls	*p*-Value
N	25	8	22	
Male sex, n (%)	17 (68.0)	3 (37.5)	14 (63.6)	0.295
Age (years), median (IQR)	50 (18, 46)	50 (24, 15)	50 (23, 76)	0.403
Smokers, n (%)	2 (8.0)	2 (25.0)	1 (4.5)	0.219
* Ex-smokers, n (%)	5 (20.0)	2 (25.0)	5 (22.7)	0.948
Subjects with other comorbidities, n (%)	10 (40.0)	5 (62.5)	8 (36.4)	0.425

* Subjects who had stopped smoking more than 12 months earlier.

## Data Availability

Anonymized data and code cannot be uploaded to a public repository, but remain available upon reasonable request to the lead contact, prof. Laura Capelli (laura.capelli@polimi.it).

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
