# Peer review of "An Experimental Apparatus for E-Nose Breath Analysis in Respiratory Failure Patients"

_diagnostics, 2022, doi:10.3390/diagnostics12040776_

Round 1
Reviewer 1 Report
The Authors presented very interesting studies of the application of an electronic nose to the detection of respiratory disorders by patients' breath analysis.
The Authors seems to be well known in gas monitoring society and enose technology, therefore the manuscript devoted to analytical part is written simply and clearly. However, applicative part should be improved. The reported performance indicates that the proposed method allows for good discrimination between respiratory failures patients from controls.
Some aspects must be clarified if the manuscript is considered to be published in Diagnostics. Listed below critique points should be addressed before the manuscript can be considered for publication.
- The authors should prepare Graphical Abstract to make the manuscript more attractive for potential readers.
- The abstract is written in a general way, the details about the analytical parameters should be included.
- Recent abbreviations concerning electronic nose – EN should be included, accompanied with recent literature: 10.1016/j.aca.2019.05.024, 10.3390/s18041052
- The Authors should revised manuscript according to Guide for authors attached in journal (e.g. Tables, figures captions, etc.). Also, all Latin phrases (via, i.e., e.g., in-situ, et al., etc…) in scientific writing should be in italics and abbreviations should be explained while using for the first time. Moreover, scientific writing rules concerning units spacing should be included: https://physics.nist.gov/cuu/Units/checklist.html and journal internal instruction
- Line 36, literature accompanied to clinical test applications should be presented.
- Line 49, I believe that the correct statement is odour or smell fingerprint.
- Line 54, several other applications of ENs should be included, e.g. food/pharmaceutical industry.
- What tests are routinely performed in laboratories and what tests are performed as part of various trials? What standardized analytical methods are used for this? It is important to present an accurate image of the current status of medical diagnostics based on VOC analysis in order to emphasize what are the opportunities for ENs to contribute to this field.
- In the Introduction, a clear presentation of the current limitations of ENs in comparison with more established techniques is missing. In general all is exposed in broad general terms without much quantitative information.
- Real view of breath sampling apparatus or photo taken during sampling should be added to Fig 1. Also, I believe that picture of sampling bag is not bringing additional value to the manuscript. The Authors can use Enose with connected bag instead of bag alone. Besides, Why Nalophan bags were used? Tedlar bags are much more popular and easier to handle. Is it a price?
- It is not precise in the paper but it seems that only one measurement of each patient was performed. It should be explicitly written. Did the Authors perform only one repetition of measurements for each patient?
- The collection of data took several months from March until September 2021. Do the patients belonging to three considered categories were measured in random order? Since there were 55 patients, with one measurement for each, it could happen that during such a long period the sensors drift is important order of measurement may be an important bias.
- Line 233, paragraph feasibility of experimental set-up... should be expanded.
- Figures 5, 6, 7 should be merged.
- The significance and novelty of the presented approach is not clearly stated. A table for the comparison of the proposed system with the similar measuring technology from commercial and/or literature should be given.
Author Response
1
We would like to thank the Reviewer for the time she/he has dedicated to revise our manuscript. We
are also very glad that our research has been appreciated. We did our best to respond to all the
Reviewer’s comments and we hope that we have addressed all her/his perplexities adequately.
In particular,
Comments:
The Authors presented very interesting studies of the application of an electronic nose to the detection
of respiratory disorders by patients' breath analysis. The Authors seems to be well known in gas
monitoring society and enose technology, therefore the manuscript devoted to analytical part is
written simply and clearly. However, applicative part should be improved. The reported performance
indicates that the proposed method allows for good discrimination between respiratory failures
patients from controls. Some aspects must be clarified if the manuscript is considered to be published
in Diagnostics. Listed below critique points should be addressed before the manuscript can be
considered for publication.
Q. 1
The authors should prepare Graphical Abstract to make the manuscript more attractive for potential
readers.
Response:
We prepared a graphical abstract as suggested by the Reviewer.
Q. 2
The abstract is written in a general way, the details about the analytical parameters should be
included.
Response:
We have revised the abstract accordingly.
Q. 3
Recent abbreviations concerning electronic nose – EN should be included, accompanied with recent
literature: 10.1016/j.aca.2019.05.024, 10.3390/s18041052.
2
Response:
We thank the author for this suggestion. We used the expression “EN” as abbreviation of “electronic
nose”, and we added the suggested references in the text (ref. 19 and 22)
Q. 4
The Authors should revised manuscript according to Guide for authors attached in journal (e.g.
Tables, figures captions, etc.). Also, all Latin phrases (via, i.e., e.g., in-situ, et al., etc...) in scientific
writing should be in italics and abbreviations should be explained while using for the first time.
Moreover, scientific writing rules concerning units spacing should be included:
https://physics.nist.gov/cuu/Units/checklist.html and journal internal instruction.
Response:
We applied the indication from the Guide for authors, we have written Latin abbreviations in italics
throughout the manuscript and formatted the units as per the provided instructions.
Q. 5
Line 36, literature accompanied to clinical test applications should be presented.
Response:
We added the reference [1]
Q. 6
Line 49, I believe that the correct statement is odour or smell fingerprint.
Response:
We changed the statement from “smell print” into “odour fingerprint” throughout the manuscript, as
suggested.
Q. 7
Line 54, several other applications of ENs should be included, e.g. food/pharmaceutical industry.
Response:
We acknowledge that ENs have been widely used in several applications, including
food/pharmaceutical industry, environmental analysis and so on. However, here we prefer to focus
our introduction on the use of ENs for diagnostic purposes, in order to avoid writing a too long and
dispersive introduction. Nonetheless, in order to meet the Reviewer’s request, we tried to better
specify our focus as follows: “Indeed, there are several scientific studies discussing the possibility to
diagnose different diseases from the analysis and the identification of volatile biomarkers in the
exhaled breath by means of different analytical techniques. Most studies focus on the non-invasive
detection of cancer, but there are also other examples regarding tuberculosis and liver cirrhosis.
Reviewer 2 Report
This is an interesting paper that has developed a means of lung washout followed by bag collection. The paper is generally well written and presented, though the authors seems to be a little behind on current Covid-19 breath papers – this will need to be addressed. I am concerned about the statistics and it maybe that the authors have simple not explained it well enough. Otherwise, this will need to be redone to be scientifically sound. I would also like to see more details on the instrument setup and the sensors. Some comments are below.
Line 44: This paragraph should be referenced.
Line 45: It can contain upto 3000, not everyone has 3000 chemicals in their breath. I would also say “measured” rather than “sampled”.
Line 52: These are old references and there has a been a lot of work since this point. You should update your references and expand on previous work.
Line 60: A G.A.S. BreathSpec is pretty close and you just take a room air and subtract. It was designed for patients with limited respiratory function, so I don’t fully agree with this statement. There are a number of other devices that use room air measurements to remove background VOCs, so please update this section.
Line 66: There are a number of papers on Covid-19 and breath analysis with respiratory disease. These should be referenced.
Line 82: So, like the Owlstone ReCIVA?
Line 84: Fix reference
Line 98: Please add volume of the bags.
Line 125: How was this checked?
Line 154: How long did it take to fill the bag and roughly the range of breaths needed?
Line 163: Please can you add details of these custom MOX sensors. Also, please can you provide details of how the instrument was set up.
Line 183: I have a serious concern here. I have no problems with your feature extraction approach, but feature selection should have been done inside the 10-fold cross validation. In your flow chart you are using all the data to select the best features beforehand. This will need to be re-done or it might be the figure is wrong.
Line 234: Was the patients tested at random or one group first or? Also please fix the English.
Line 293: You have done a ROC, so please can you provide sensitivity/specificity/p-value and confidence intervals for the ROC and the rest. You could also provide accuracy if you have it.
Author Response
We would like to thank the Reviewer for the time she/he has dedicated to revise our manuscript. Weare also very glad that our research has been appreciated. We did our best to respond to all the
Reviewer’s comments and we hope that we have addressed all her/his perplexities adequately.
In particular,
Comments:
This is an interesting paper that has developed a means of lung washout followed by bag collection.
The paper is generally well written and presented, though the authors seems to be a little behind on
current Covid-19 breath papers – this will need to be addressed. I am concerned about the statistics
and it maybe that the authors have simple not explained it well enough. Otherwise, this will need to
be redone to be scientifically sound. I would also like to see more details on the instrument setup and
the sensors. Some comments are below.
Q. 1
Line 44: This paragraph should be referenced.
Response:
Thanks for the comment. We added a reference [2] to the paragraph.
Q. 2
Line 45: It can contain up to 3000, not everyone has 3000 chemicals in their breath. I would also say
“measured” rather than “sampled”.
Response:
We thank the reviewer for this comment that gave us the possibility to improve the quality of the
manuscript. We agree and added the ‘up to’ concept. Moreover, we added references
https://doi.org/10.1007/s00408-017-9987-3, 10.1088/1752-7163/ac4916 supporting this
consideration: exhaled breath samples may have more than 3000 VOCs.
Furthermore, we changed the sentence stated by the reviewer according to his/her suggestion.
2
Q. 3
Line 52: These are old references and there has a been a lot of work since this point. You should
update your references and expand on previous work.
Response:
We thank the reviewer for this comment. We updated the references by including ref [22-29].
Q. 4
Line 60: A G.A.S. BreathSpec is pretty close and you just take a room air and subtract. It was designed
for patients with limited respiratory function, so I don’t fully agree with this statement. There are a
number of other devices that use room air measurements to remove background VOCs, so please
update this section.
Response:
We verified the characteristics of G.A.S. BreathSpec and found that it is slightly different from the
system we propose. First, the exhaled breath is analysed by GC-IMS, while our system is based on
EN technology that results in different advantages (e.g., cheaper, fast response). Regarding the fact
that that the system is designed for patients with limited respiratory functions, we couldn’t find any
reference mentioning its use in respiratory failure patients, so we have no elements for highlighting
the differences between the two systems.
In order to fulfil the Reviewer’s request, we mentioned other EN devices proposed in the scientific
literature, which compensate for background VOCs by filtering ambient air during lung washout or
by executing blank tests in parallel with breath analysis (lines 78-83).
Q. 5
Line 66: There are a number of papers on Covid-19 and breath analysis with respiratory disease.
These should be referenced.
Response:
We included in the Introduction refences to papers proposing the EN as a diagnostic tool for Covid-
19 (ref 33 to 36, line 106).
Q. 6
Line 82: So, like the Owlstone ReCIVA?
Response:
We thank the reviewer for this comment. Owlstone ReCIVA presents several differences from the
instrument proposed in our study. First, VOCs are captured in cartridges and then analysed in a second
moment, while in this application VOCs are analysed directly in the gas phase. Second, even if
Owlstone ReCIVA presents a low mechanical load, apparently it does not have the possibility to filter
out external VOCs or to use medical grade air with known oxygen concentration. In our opinion,
3
those aspects represent very important features for a device to be used with patients with respiratory
failure.
Q. 7
Line 84: Fix reference.
Response:
There was no reference in that point, we cancelled the error message.
Q. 8
Line 98: Please add volume of the bags.
Response:
We used Nalophan bag with a capacity of 5L. We added this information in line 132.
Q. 9
Line 125: How was this checked?
Response:
We thank the reviewer for this comment. We added details in the mentioned section, modifying the
text as follow:
Clinicians asked to the subject to stop the measurement if they felt discomfort or dispnea, and
saturation was measured before, during and after the measurement
Q. 10
Line 154: How long did it take to fill the bag and roughly the range of breaths needed?
Response:
We added in Section 2.2.2. a sentence specifying the duration of the breath sampling and the number
of patient’s breaths needed (lines 183-184).
Q. 11
Line 163: Please can you add details of these custom MOX sensors. Also, please can you provide
details of how the instrument was set up.
Response:
4
We thank the Reviewer for the comment. We implemented the Section 2.2.3. with information about
the type of MOS sensor used and the temperature at which they were operated for the analyses of
breath samples (lines 195-197).
Q. 12
Line 183: I have a serious concern here. I have no problems with your feature extraction approach,
but feature selection should have been done inside the 10-fold cross validation. In your flow chart
you are using all the data to select the best features beforehand. This will need to be re-done or it
might be the figure is wrong.
Response:
The feature selection has been actually implemented inside the 10-Fold Cross Validation, and we
want to thank the reviewer for his comment that allowed us to realize that probably this procedure
wasn’t explained clearly enough. Therefore, we tried to add an explanation in lines 213-215 and 248-
249 and we replaced the Figure 3, aiming to clarify the procedure adopted.
Q. 13
Line 234: Was the patients tested at random or one group first or? Also please fix the English
Response: We thank the reviewer for the question. Patients were tested in random order. We
reformulated the sentence to improve its clarity (line 267).
Characteristics of study participants
A total of 55 subjects were recruited and tested in random order
Q. 14
Line 293: You have done a ROC, so please can you provide sensitivity/specificity/p-value and
confidence intervals for the ROC and the rest. You could also provide accuracy if you have it.
Response:
We implemented the Section 3.4., adding the information about sensitivity, specificity and accuracy
(lines 338 - 343).
Round 2
Reviewer 1 Report
Accept
Author Response
We thank the Reviewer for the time he/she dedicated to revise again our paper.
Reviewer 2 Report
Thank you for your improvements of the paper. Please can you just add the confidence intervals to the AUC, Sensitivity, Specificity etc. and then I will be happy with it.
Author Response
We thank the Reviewer for the time he/she dedicated to revise again our paper.
According to Reviewer's suggestion, we added the confidence intervals of AUC, Sensitivity and Specificity.
Moreover, we have revised the Enghish throgouth the manuscript.